# The Major Components of Cerebrospinal Fluid Dictate the Characteristics of Inhibitors against Amyloid-Beta Aggregation

**DOI:** 10.3390/ijms24065991

**Published:** 2023-03-22

**Authors:** Andrius Sakalauskas, Mantas Ziaunys, Ruta Snieckute, Agne Janoniene, Dominykas Veiveris, Mantas Zvirblis, Virginija Dudutiene, Vytautas Smirnovas

**Affiliations:** Institute of Biotechnology, Life Sciences Center, Vilnius University, LT-10257 Vilnius, Lithuania

**Keywords:** protein aggregation, amyloid-beta, inhibition, EGCG, VR16-09, aggregation kinetics, near-physiological conditions

## Abstract

The main pathological hallmark of Alzheimer’s disease (AD) is the aggregation of amyloid-β into amyloid fibrils, leading to a neurodegeneration cascade. The current medications are far from sufficient to prevent the onset of the disease, hence requiring more research to find new alternative drugs for curing AD. In vitro inhibition experiments are one of the primary tools in testing whether a molecule may be potent to impede the aggregation of amyloid-beta peptide (Aβ_42_). However, kinetic experiments in vitro do not match the mechanism found when aggregating Aβ_42_ in cerebrospinal fluid. The different aggregation mechanisms and the composition of the reaction mixtures may also impact the characteristics of the inhibitor molecules. For this reason, altering the reaction mixture to resemble components found in cerebrospinal fluid (CSF) is critical to partially compensate for the mismatch between the inhibition experiments in vivo and in vitro. In this study, we used an artificial cerebrospinal fluid that contained the major components found in CSF and performed Aβ_42_ aggregation inhibition studies using oxidized epigallocatechin-3-gallate (EGCG) and fluorinated benzenesulfonamide VR16-09. This led to a discovery of a complete turnaround of their inhibitory characteristics, rendering EGCG ineffective while significantly improving the efficacy of VR16-09. HSA was the main contributor in the mixture that significantly increased the anti-amyloid characteristics of VR16-09.

## 1. Introduction

Alzheimer’s disease (AD) belongs to a group of disorders identified as neurodegenerative diseases [1]. The main pathophysiological hallmark of AD is the accumulation of extracellular senile plaques (SPs) and neurofibrillary tangles [2]. Amyloid β (Aβ) is considered the main component of these plaques and influences the onset of AD [3]. The constant production of 40-amino acid-long Aβ (Aβ_40_) by proteolysis of the integral membrane protein called amyloid precursor protein (APP) is a normal process that occurs throughout the entire lifespan of humans [4]. However, the dysregulation or an incorrect proteolytic cleavage of APP, producing 42-amino acid-long Aβ (Aβ_42_), may lead to a continuous imbalance between the production and elimination of the peptide, resulting in elevated levels of Aβ_40_ and Aβ_42_ [5]. This cascade step is followed by the oligomerization and fibrillation of Aβ that drives the accumulation of SPs [6].

Various therapies aim to reduce or prevent the progression of AD [7]. While the efficiency of these established programs is minimal, the development of drugs against this neurodegenerative disorder is carried out through different strategies aimed at one or more targets, including Aβ (production, anti-aggregation, and immunotherapy), Tau protein (immunotherapy, phosphorylation, and aggregation), neuroinflammation, and ease of the behavioral, psychological symptoms of dementia [8,9].

One of the strategies is to search for an active anti-amyloid compound that would interact with Aβ_42_ and prevent or slow the aggregation process. The primary step for this approach is screening the potential molecules in silico and in vitro [10]. In vitro screening experiments using different origins of Aβ_42_ peptides (synthetic or recombinant) found that specific reaction mixtures may lead to a distinct aggregation mechanism that could affect the inhibition studies [11,12]. First, there are studies where inhibition of Aβ_42_ was performed in a purification buffer after the final chromatographic step. In this case, the reaction mixture contained elevated pH (8.0–8.5) and reduced ionic strength compared to physiological conditions, resulting in aggregates with different morphologies and secondary structures [13]. Second, synthetic Aβ_42_ peptide aggregates slower than the recombinant version, affecting the aggregation pathway and cell toxicity [12]. In addition, a prolonged aggregation may lead to secondary events, such as degradation and modification of the inhibitor or the fluorescent dye used in the aggregation assay [14], that could shift the perception of the inhibition process. Finally, PBS is the most common buffered system where Aβ_42_ is aggregated. While the medium possesses similar pH and ionic strength, it lacks the main components typically found in cerebrospinal fluid (CSF) [15].

CSF surrounds the brain tissue, and Aβ_42_ aggregation may occur in the drainage of brain interstitial fluid [16]. CSF is a clear, colorless fluid and contains more sodium ions, chloride ions, and glucose but less calcium and potassium ions than in the blood [17]. Aβ_42_ assembly into fibrillar aggregates differs in CSF than in PBS or Hepes buffered systems [18,19]. It is known that both non-chaperone [20,21,22] and chaperone [23] proteins affect the aggregation of Aβ in vivo and in vitro. One of these proteins is albumin, found in CSF (<0.4 mg/mL), and is up to 90% of the total protein count in the fluid [24]. Stanyon and Viles showed that human serum albumin (HSA) regulates the aggregation of Aβ_42_ by binding to the peptide and trapping it in a nonfibrillar form [25]. In addition to HSA, transthyretin [26], transferrin [27], immunoglobulins, and other proteins are found in nanomolar concentrations [24]. Furthermore, CSF contains amino acids that may also contribute to altering the aggregation. According to research by Rainesalo, glutamine (450 μM–900 μM) was the main amino acid found in the CSF (control subjects). In contrast, the concentration of all other amino acids was up to 25 μM [28].

Although it is impossible to recreate CSF perfectly to contain the equivalent levels of the vast number of components, it is necessary to improve the search for anti-amyloid compounds in vitro by mimicking the composition of CSF. Specific molecules present in CSF, such as albumin [20] or metal ions [29,30], may alter the aggregation pathway, which could completely change the properties of the inhibitor. The composition of artificial cerebrospinal fluid (aCSF) should minimize the barrier between the results in vitro and in vivo.

In this work, we have created an aCSF containing major components of CSF and tested how it affects the Aβ_42_ aggregation process. Two inhibitors were used to inhibit Aβ_42_ aggregation in PBS and the aCSF. One is fluorinated benzenesulfonamide VR16-09, previously shown to possess anti-amyloid characteristics against Aβ, α-synuclein, and insulin aggregation [31]. The other inhibitor used in this study was oxidized EGCG. EGCG was found to be unstable and prone to autoxidize at neutral or higher pH, which enhances its potency to inhibit various protein aggregation, including Aβ and α-synuclein [32,33]. We show that both inhibitors acted differently in the selected solutions, altering the conclusions drawn about their anti-amyloid properties.

## 2. Results

Aβ_42_ and its aggregates are found in the cerebrospinal fluid surrounding brain tissues [3]. This is why it is essential to alter the in vitro screening conditions to closely match the main components of the cerebrospinal fluid in order to gain confidence in the results. The created aCSF used in the experiments contained glucose, CaCl_2_, MgCl_2_, glutamine, urea, cholesterol, lactate, and human serum albumin.

In this work, we compared Aβ_42_ aggregation in PBS and aCSF (Figure 1). The two chemically different inhibitors, fluorinated benzenesulfonamide VR16-09 (Appendix A) [31] and oxidized polyphenol EGCG (Appendix A) [32], were used to account for the aggregation inhibition changes among the selected solutions. In PBS (Figure 1A), oxidized EGCG increased Aβ_42_ aggregation halftime two-fold and reduced the final ThT fluorescence intensity. This intensity reduction was previously shown to be related to ThT fluorescence quenching due to its interaction with the oxidated polyphenols or inner filter effects [34,35]. VR16-09, on the other hand, proved to be less impactful, with almost no reduction in the aggregation halftime compared with the control sample. However, there was a significantly higher baseline of VR16-09 than oxidized EGCG (intrinsic fluorescence in the selected emission wavelength range). Fluorescence intensity curves of inhibitor compounds in the absence of Aβ_42_ are presented in Appendix A. A completely opposite result was observed when Aβ_42_ aggregated in the aCSF (Figure 1B). While oxidized EGCG reduced the final ThT fluorescence intensity level, the aggregation curves reached a plateau earlier than the control sample. In the sample where Aβ_42_ aggregated with VR16-09, the aggregation halftime was increased with no shift of the ThT fluorescence intensity. It is important to note that in both reaction solutions, Aβ_42_ aggregation halftime with oxidized EGCG was of a similar value (Figure 1C), while VR16-09 had a complete turnover, significantly slowing Aβ_42_ aggregation in the aCSF reaction mixture. This shift, however, was not observed during the MTT assay in cells (Appendix A).

Further, it was decided to compare the morphology of Aβ_42_ aggregates formed in PBS and aCSF using AFM (Figure 2A,D). The Aβ fibrils in PBS and aCSF were mostly clumped together into larger structures with a limited number of single fibrils. The fibril height distribution of control samples (Figure 2G) showed no significant difference and correlated to fibril heights found in the literature [36]. A similar view was seen in AFM images of samples where Aβ_42_ was aggregated with oxidized EGCG (Figure 2B,E). However, in the case of aCSF, a vast number of smaller, round-shape structures was observed, while in PBS, the structures were less abundant and larger. Despite this fact, there was no significant difference in their height distribution (Figure 2H). In contrast, Aβ_42_ aggregates formed with VR16-09 in PBS were shorter and significantly higher compared to the fibrils formed in the aCSF (Figure 2C,F,I). Additionally, Aβ fibrils formed in the aCSF with VR16-09 were longer than in the sample with oxidized EGCG.

The aCSF is a more complex reaction mixture than PBS, containing glucose, Ca^2+^, Mg^2+^ ions, urea, cholesterol, lactate, glutamine, and HSA. For this reason, experiments were conducted where each of the aforementioned components was used in Aβ_42_ aggregation assays with and without VR16-09 (Figure 3). CaCl_2_, MgCl_2_, and urea sped up the Aβ_42_ aggregation, while glutamine and HSA contributed to a longer fibrillation time (Figure 3H). Cholesterol, lactate, and glucose did not significantly affect the assembly of Aβ_42_ fibrils. The Aβ_42_ aggregation inhibition using VR16-09 was not achieved in the reaction without additional aCSF components, where only NaCl, KCl, KH_2_PO_4_, Na_2_HPO_4_, and NaH_2_PO_4_ components were present (Figure 3A) (Further referred to as w/o aCSF components). In the presence of any component, the inhibition of VR16-09 was altered. All the components, except glutamine, enhanced the inhibitory effect of VR16-09. The most impactful element was HSA (Figure 3D), increasing both to Aβ_42_ aggregation halftime and the inhibitory effect of VR16-09 more than two-fold. On the other hand, glutamine negatively contributed to the inhibitory potential of VR16-09. When all the components were present (Figure 3G), the inhibitory effect was increased, although the aggregation halftime of the control sample was reduced compared to the sample with HSA.

AFM images revealed that Aβ_42_ aggregates formed without additional aCSF components (Figure 3B,I) were up to several micrometers in length and 2–6 nm in height while being irregularly clumped together or dispersed throughout the mica. When VR16-09 was present in the sample (Figure 3C), fibrils were of similar length but more clumped together, forming larger structures. Even larger clusters of aggregates were found when Aβ_42_ was aggregated with HSA (Figure 3E). However, the samples had visually longer fibrils than in the previous cases. When HSA and VR16-09 were present in the reaction mixture (Figure 3F), fewer fibrils were found on the mica. They were up to 2–3 µm in length and 3–8 nm in height. No larger clumps of aggregates were found in this particular case. An interesting instance to note, the addition of VR16-09 significantly increased the height of the fibrils, but it did not contribute to any significant height difference among the samples with the inhibitor, regardless of whether HSA was present or not.

To further determine the impact of VR16-09 on the aggregation of Aβ_42_ in aCSF, Aβ_42_ was aggregated under a range of protein concentrations from 0.75 to 2.0 μM, with or without 25 μM of VR16-09 (Figure 4A,B). To compare the inhibition, Aβ_42_ was also aggregated in aCSF without serum albumin (Figure 4C,D), which seemed to make the greatest impact on the inhibitory effect of VR16-09 (Figure 3) as well as on the morphology of the fibrils formed and Aβ_42_ toxic impact on cells during MTT assay (Appendix A). Instead of measuring the aggregation halftime, global fitting of the data was performed using a four-step model (nucleation, elongation, secondary nucleation, and fragmentation), as described previously [37]. The data were corrected for the initial signal intensity drop to allow for a more accurate fitting procedure. The entire procedure is described in the Materials and Method section. Fit aided in calculating combined rate constants of primary nucleation–elongation (Figure 4E), elongation–secondary nucleation (Figure 4F), and elongation–fragmentation (Figure 4G). The kinetic and global-fitting data suggested that the addition of albumin and VR16-09 prolongs the aggregation of Aβ_42_; however, in a different pattern. Removal of HSA from the aCSF formulation, whether VR16-09 was present or not, contributed to a sizeable decrease in the combined elongation–fragmentation rate constant (Figure 4G) while revealing an increase in primary nucleation–elongation and elongation-secondary nucleation (Figure 4E,F) rate constants in the environment with the VR16-09 compound. Accordingly, a lower fragmentation was visible in the previously shown AFM images (Figure 3).

On the other hand, VR16-09 had a profound effect on the combined rate constants by significantly decreasing primary nucleation–elongation and considerably reducing elongation–secondary nucleation, while the elongation–fragmentation constant was unaffected. Specifically, this trend was visible to the naked eye in aCSF when VR16-09 was added. The Aβ_42_ aggregation lag time increased, but the slope of the kinetic curves remained similar.

## 3. Discussion

Perhaps because the initial screening experiments when searching for the anti-amyloid compound were initiated in PBS or other reaction mixtures used to aggregate Aβ_42_ (not in CSF), the components of cerebrospinal fluid were overlooked as crucial contributors to the inhibition studies. Out of eight compounds chosen in the formulation of the aCSF (Figure 3), HSA and glutamine had an inhibitory effect against Aβ_42_ aggregation. While the inhibitory effect of HSA is known to be a “Monomer-competitor” [20], there is no record of glutamine affecting the fibrillation process of Aβ_42_. This amino acid may interact with Aβ_42_ through the formation of hydrogen bonds, hence stabilizing the protein. Compared to studies with inhibitors [31,34,38], a 0.7 mM concentration would be considered very high. Such inhibitor levels should be avoided due to potential toxic side effects on cells. However, the glutamine levels in CSF of regular patients can be even higher (0.9 mM) [28], enabling the possible regulation of Aβ_42_ aggregation. In addition to the inhibitors present in the aCSF (Figure 3), the addition of Ca^2+^ and Mg^2+^ sped up the aggregation. This correlates with previous observations, as it has been shown that Ca^2+^ promotes the oligomerization of intracellular Aβ_42_ [29].

The more complex picture was when all the components were present in the reaction mixture. The aggregation of Aβ_42_ in the aCSF was enhanced by Ca^2+^, Mg^2+^ and inhibited by HSA and glutamine. Unfortunately, the situation became less clear because the components in the mixture were likely prone to cross-interactions. For example, Bode et al. showed that cholesterol, fatty acids, and warfarin suppress the inhibitory effect of HSA on Aβ_42_ [39]. Both of these components were present in aCSF. However, Aβ_42_ aggregation halftime was more than 2-fold higher in the aCSF compared to PBS but lower than in the reaction mixture with HSA only (Figure 3A,D,G). A similar trend was observed by Padayachee et al., who compared Aβ_42_ aggregation in a Hepes buffered system and in CSF samples from human patients [18]. This could mean a close relationship between the components in CSF and aCSF, affecting Aβ_42_ aggregation. Nonetheless, the link between the components and the aggregation mechanism of Aβ_42_ remains convoluted.

In PBS, in the presence of oxidized EGCG, the Aβ_42_ fibrillation process slowed (Figure 1A). As it is typical to polyphenolic molecules containing neighboring hydroxy groups, the oxidation of EGCG may lead to the formation of polymeric molecules that are capable of binding to lysine and arginine amine groups that stabilize the protein monomer [32,40,41]. This would help to explain the accelerated fibrillation process in aCSF (Figure 1B). HSA with both lysine and arginine groups was present in the reaction mixture, which may bind to and reduce the number of oxidized EGCG molecules that may covalently interact with Aβ_42_. This hypothesis is supported by AFM images (Figure 2). A number of smaller spherical objects were seen in the sample of aCSF, while in PBS, the structures were larger and less abundant. In the case of VR16-09, the opposite effect was observed. PBS had no inhibitory effect, possibly due to the solubility issues. However, matters changed when Aβ_42_ was aggregated in aCSF, where the inhibitory effect of VR16-09 was greatly enhanced. This could be explained by the combined inhibition reaction with the components of aCSF. The claim is supported by the results presented in Figure 3, where all the components, except glutamine, enhanced the inhibitory effect. This also explains why Aβ_42_ aggregation without any aCSF components was not inhibited by VR16-09 (Figure 3A). This result, however, contradicts the previously published data [31]. This inconsistency may have appeared due to several layers of mismatch—the nature of the Aβ_42_ peptide (synthetic or recombinant), the concentration of the inhibitor, and the difference in preparation procedures.

The experiments in aCSF revealed its profound effect on the potency of amyloid inhibitors. VR16-09 was previously introduced as an insulin aggregation inhibitor with only low-to-moderate applicability against Aβ_42_ aggregation [31] (at higher inhibitor concentrations). This fluorinated benzenesulfonamide showed a pronounced effect and inhibited Aβ_42_ aggregation in aCSF by suppressing primary nucleation (Figure 4), associated with an extended aggregation lag time [42]. The interesting aspect of this inhibition is that VR16-09 and HSA had a combined effect by suppressing primary nucleation. This could be related to the interaction between VR16-09 and Aβ_42_ that may reduce the number of free monomers in the sample capable of forming a nucleus. Further, it is possible that during Aβ_42_ aggregation, VR16-09 binds to the fibrils, accelerating the surface catalysis reflected in an increased secondary nucleation rate. When HSA was present, this effect may be countered by decreasing affinity between Aβ_42_ monomers and its fibrils’ surface or by reducing the number of free monomers in the sample [20]. While VR16-09 was not the most potent, the need for screening in an altered medium consisting of molecules found in CSF that affect Aβ_42_ aggregation is critical. Molecules, such as EGCG, that did not pass clinical trials were thought to lack the oxidation step, which is a key factor enabling its inhibition against protein aggregation [32]. However, after analyzing the results presented in this manuscript, the oxidized EGCG did not show any inhibitory effect against Aβ_42_ aggregation in aCSF, except for the reduced ThT fluorescence intensity, which can be altered by the presence of exogenous compounds [43].

## 4. Materials and Methods

### 4.1. Preparation of aCSF Reaction Mixture

The artificial cerebrospinal fluid was prepared based on the findings of human cerebrospinal fluid [24,44,45] and previously published compositions of aCSF [28,46,47] to contain the main components that may interact with protein aggregation and inhibition processes. A concentrated aCSF solution was prepared in three separate parts, listed in Table 1. The three parts were selected in order to avoid Ca^2+^ and Mg^2+^ ion reactivity with phosphate when dissolving them. HSA, cholesterol, sodium lactate, and glutamine (Part 3) were chosen to be dissolved separately to prevent the impact of high salt concentration. To prepare 1× aCSF, three parts were mixed together and diluted using MilliQ water at the ratio 9:9:10:72 (Part 1:Part 2:Part 3:MilliQ water). To account for the pH difference between the prepared solutions and the one measured in CSF (7.33), 1.6 µL (*v*/*v*) of 25% HCl was added to 10 mL of 1× aCSF, which yielded a pH of 7.33. 100× D-glucose, CaCl2, MgCl2, Urea, sodium lactate, 10× HSA, cholesterol, and glutamine stocks were prepared for the experiments where separate components were used.

### 4.2. Preparation of Epigallocatechin-3-Gallate (EGCG)

EGCG is prone to autoxidation at neutral or higher pH resulting in the formation of different autoxidation products. These products are shown to possess an elevated inhibitory effect against protein aggregation. In order to avoid EGCG autoxidation during the protein aggregation experiments that may render the process unstable, EGCG was fully autoxidized. EGCG powder (Fluorochem, Glossop, UK, cat. No. M01719) was dissolved in 100 mM potassium phosphate buffer, pH 7.4, to a final concentration of 10 mM. Then, 1 mL of the solution was placed into 1.5 mL test tubes and incubated for 72 h at 60 °C. After this procedure, the test tubes were stored at −20 °C.

### 4.3. Aβ_42_ Aggregation Experiments

The Aβ_42_ peptide was expressed in *E. coli* BL-21StarTM (DE3) (Invitrogen, Carlsbad, CA, USA) and purified using the expression vector, as described previously [34,48]. The purified peptide fraction (8–20 μM) (1.5 mL, 20 mM sodium phosphate, 0.2 mM EDTA pH 8) was mixed together with 10× PBS or concentrated aCSF parts and 10 mM thioflavin-T (ThT) stock solutions (Sigma-Aldrich, St. Louis, MO, USA, cat. No. T3516), MilliQ water, and 10 mM oxidized EGCG or 10 mM VR16-09 stock solutions (VR16-09 stock solution was prepared as previously described [31]) to yield 2 µM Aβ_42_ and 20 µM ThT in PBS or aCSF with or without 25 µM of the corresponding inhibitor. For experiments where a range of Aβ_42_ concentrations was used, the prepared solution (2 µM Aβ_42_, 20 µM ThT in aCSF) was diluted using a reaction solution (without Aβ_42_) and 10 mM VR16-09 stock solution (dissolved in DMSO) to yield 2, 1.5, 1.25, 1.0, 0.75 µM of Aβ42, and 20 µM ThT with or without 25 µM VR16-09. The equivalent volume of DMSO was used for the control samples. Kinetic aggregation measurements were performed in Corning non-binding 96-well plates (Fisher, Waltham, MA, USA, cat. No. 10438082) (sample volume was 80 µL) at 37 °C by measuring ThT fluorescence, using 440 nm excitation and 480 emission wavelengths, in a ClarioStar Plus plate reader (BMG Labtech, Ortenberg, Germany).

### 4.4. Atomic Force Microscopy (AFM)

The samples for AFM images were taken after Aβ_42_ kinetic aggregation measurements were completed and scanned similarly, as previously described [34]. In brief, 40 µL of 0.5% (*v*/*v*) APTES (Sigma-Aldrich, cat. No. 440140) in MilliQ water was distributed onto freshly cleaved mica to modify the surface in order to bind the negatively charged structures. After incubating for 5 min, the mica was washed with 2 mL of MilliQ water and then dried under modest airflow. Forty microliters of each sample was deposited on the functionalized surface, incubated for another 5 min, washed with 2 mL MilliQ water, and dried under airflow. Imaging was done using a Dimension Icon (Bruker, Billerica, MA, USA) atomic force microscope. One thousand and twenty-four- by one thousand and twenty-four-pixel resolution images were recorded using a Nanoscope 10.0 (Bruker) and analyzed using Gwyddion 2.57 software. The heights of the fibrillar structures found on the mica were determined by tracing perpendicular to each fibril axis.

### 4.5. Cell Culturing

SH-SY5Y human neuroblastoma cells were obtained from the American Type Culture Collection (ATCC, Manassas, VA, USA). The cells were grown in Dulbecco’s Modified Eagle Medium (DMEM) (Gibco, Grand Island, NY, USA), supplemented with 10% Fetal Bovine Serum (FBS) (Sigma-Aldrich, St. Louis, MO, USA), 1% Penicillin–Streptomycin (10,000 U/mL) (Gibco, Grand Island, NY, USA) at 37 °C in a humidified, 5% CO_2_ atmosphere in a CO_2_ incubator.

### 4.6. MTT Assay

MTT assay of SH-SY5Y cells was performed as previously described [40]. In short, SH-SY5Y cells were seeded in a 96-well plate (~15,000 cells/well) and cultured overnight. The 2 µM Aβ_42_ monomers with or without 25 µM oxidized EGCG or 25 µM VR16-09 in PBS or aCSF were diluted in half with DMEM and used to replace the cell medium in each well. The preformed fibrils were used for the cell metabolic activity experiment using Aβ_42_ with or without 10 µM of BSA. After the kinetic experiments, samples were taken, diluted in half with DMEM, and placed onto the cells in a 96-well plate. After 48 h of incubation, 10 µM of MTT was added to each well and left to incubate for 2 h. One hundred microliters of 10% SDS with 0.01 M HCl solution was added on top to dissolve formazan crystals. Absorbances at 540 nm, 570 nm, and 690 nm (reference wavelength) of each well were measured using a ClarioStar Plus plate reader (BMG Labtech, Ortenberg, Germany).

### 4.7. Statistical Analysis

The aggregation kinetic data analysis was done by fitting the kinetic curves using Boltzmann’s sigmoidal equation [49]. Halftime values are added in Appendix A. The relative halftime values were calculated using the control samples from their respective 96-well plates. For the experimental data fitting, the baseline signal intensity drop was corrected by subtracting the exponential fit from 6 initial points of each individual curve. Each kinetic curve was then normalized between 0 and their respective initial peptide concentration in order to prepare for the global-fitting procedure (raw data are provided as Appendix A). Data fitting using a four-step aggregation model (primary nucleation, elongation, secondary nucleation, and fragmentation) was done using rModeler (Ubicalc Software, Vilnius, Lithuania), as described previously [33]. The fibril height values of different samples (n = 50) measured from AFM images were statistically compared using the one-way analysis of variance (ANOVA). * *p* < 0.01 and ** *p* < 0.001 were accepted as statistically significant. The data were processed and analyzed using Origin software (OriginLab, Northampton, MA, USA).

## 5. Conclusions

In conclusion, Aβ_42_ aggregation was slower in aCSF than in PBS due to the combination of components found in the cerebrospinal fluid. The oxidized EGCG inhibitory effect retarded when aggregating Aβ_42_ in aCSF instead of PBS, while VR16-09 showed opposite results with significantly enhanced inhibition. Human serum albumin was the main component influencing the increased inhibitory effect of VR16-09, whereas glutamine was the only one contributing negatively to the inhibition of Aβ_42_. On the other hand, the inhibitory effect of VR16-09 on Aβ_42_ aggregation in aCSF was mainly due to a decreased combined primary nucleation–elongation and an increased elongation–secondary nucleation rate constants of the process.

## Figures and Tables

**Figure 1 ijms-24-05991-f001:**
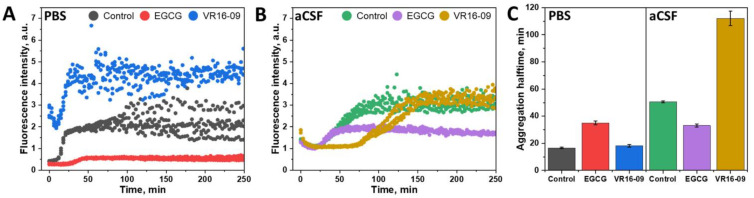
Aβ (2 µM) aggregation kinetics in PBS (**A**) and aCSF (**B**) in the absence or presence of 25 μM of oxidized EGCG or VR16-09 and their respective relative halftime values (**C**). The three separate repeats of each kinetic experiment are shown in the graphs. Error bars are one standard deviation (n = 4).

**Figure 2 ijms-24-05991-f002:**
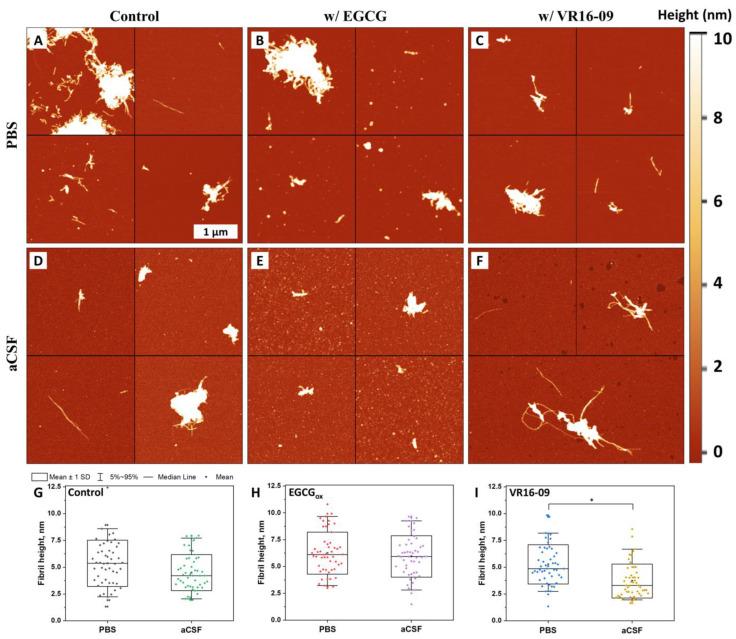
Atomic force microscopy (AFM) images of Aβ_42_ aggregates formed without (**A**,**D**) and with 25 µM of oxidized EGCG (**B**,**E**) or with 25 µM of VR16-09 (**C**,**F**) in PBS and aCSF, respectively. The fibril height distribution among aggregates formed in PBS and aCSF without (**G**) and with oxidized EGCG (**H**) or VR16-09 (**I**), where box plots indicate mean ± SD and error bars are in the 5–95% range (n = 50). ANOVA (Bonferroni) significance values were compared. * *p* < 0.01.

**Figure 3 ijms-24-05991-f003:**
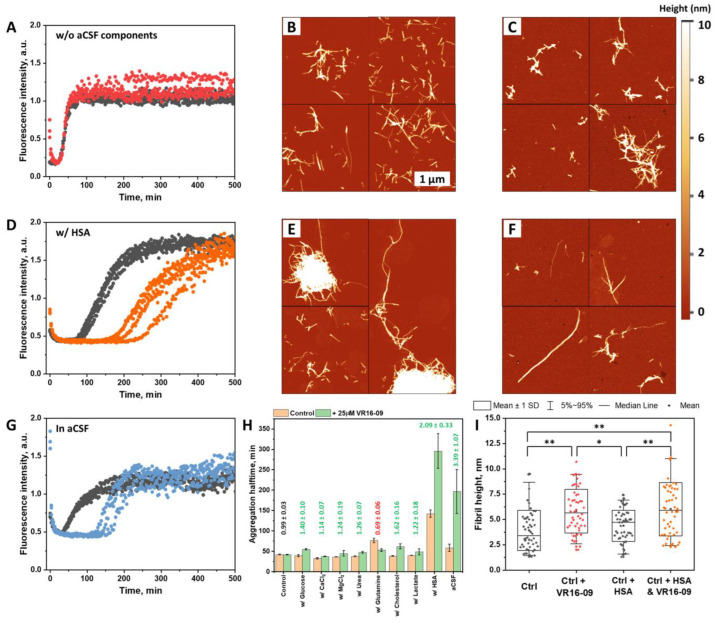
Aβ_42_ (2 µM) aggregation kinetics in the reaction mixture without aCSF components (control) (**A**), with HSA (**D**), and with all aCSF components (**G**) in the absence (black) or presence (colored) of 25 µM of VR16-09. Aβ_42_ aggregation halftime values (**H**) when aggregated with each aCSF component in the absence and presence of 25 µM of VR16-09. Relative aggregation halftime values calculated between the sample with and without the inhibitor are listed above each column in green. Error bars are one standard deviation (n = 4). Atomic force microscopy (AFM) images of Aβ_42_ aggregates formed in the reaction mixture without aCSF components (**B**,**C**) or with HSA (**E**,**F**) in the absence or presence of 25 µM of VR16-09, respectively. Their height distribution box plot (**I**) indicates mean ± SD, and error bars are in the 5–95% range (n = 50). ANOVA (Bonferroni) significance values were compared, * *p* < 0.01, ** *p* < 0.001.

**Figure 4 ijms-24-05991-f004:**
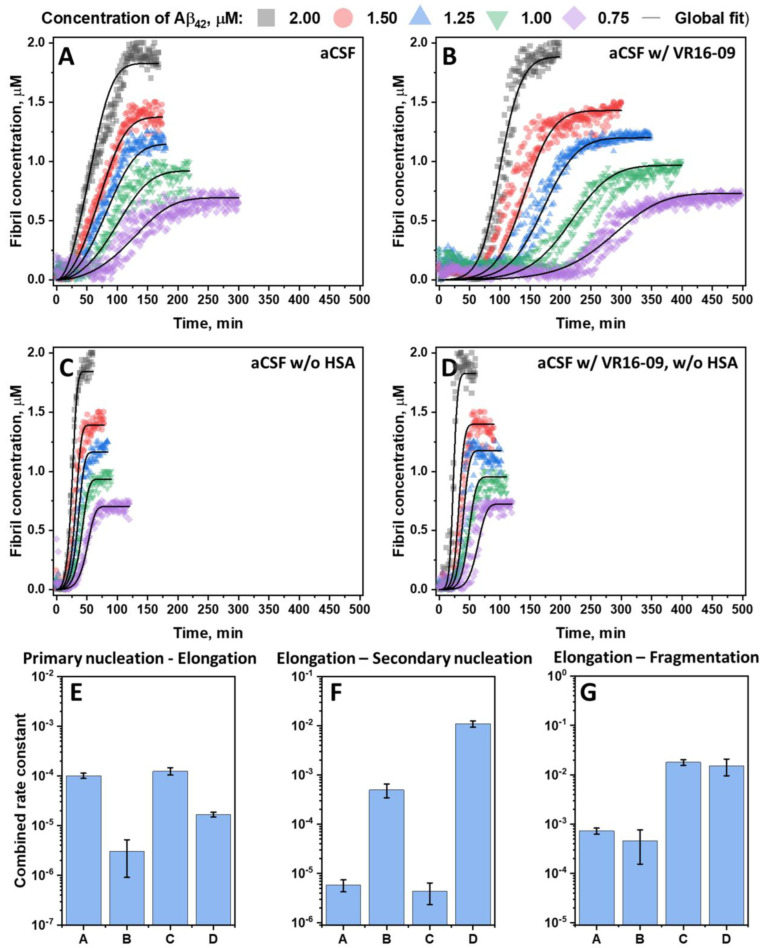
Aβ_42_ aggregation kinetics in aCSF (**A**), aCSF with 25 µM VR16-09 (**B**), aCSF without HSA (**C**), and aCSF with 25μM VR16-09 and without HSA (**D**). The fibril concentration in the *Y* axis is the concentration of monomers in their aggregated state (assuming the aggregation efficiency was 100%). For each condition, a four-step global-fitting aggregation model was used. Combined primary nucleation–elongation (**E**), elongation–secondary nucleation (**F**), and elongation–fragmentation (**G**) rate constants were obtained at each condition by global-fit of the Aβ_42_ concentration range (0.75–2 μM). Error bars are one standard deviation (n = 3).

**Table 1 ijms-24-05991-t001:** Composition of 10× aCSF.

Composition of Concentrated aCSF (in 3 Separate Parts)			
	Amount in Pt 1, mM	Amount in Pt 2, mM	Amount in Pt 3, mM
NaCl	1270	-	127
KCl	18	-	1.8
KH_2_PO_4_	12	-	1.2
Na_2_HPO_4_	78.1	-	7.81
NaH_2_PO_4_	31.9	-	3.19
D-glucose	-	40	-
CaCl_2_	-	14	-
MgCl_2_	-	13	-
Urea	-	6.5	-
HSA	-	-	0.0615
Cholesterol	-	-	0.052
Sodium lactate	-	-	24
Glutamine	-	-	7

## Data Availability

The data presented in this study are available in this manuscript.

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
