# Peer review of "The Major Components of Cerebrospinal Fluid Dictate the Characteristics of Inhibitors against Amyloid-Beta Aggregation"

_ijms, 2023, doi:10.3390/ijms24065991_

Round 1
Reviewer 1 Report
The manuscript by Sakalauskas et al. analyses the kinetics of Abeta42 peptide aggregation in the presence of two anti-amyloid model drugs, EGCG and VR16-09 in an artificial CSF environment. Many drug-screening studies have neglected the effect of CSF components both on the aggregation mechanism itself and as a potential binding partner. As this study shows, CSF components, especially albumin can alter aggregation kinetics significantly and also have a major impact on the drug effects. The study therefore highlights an important consideration in studying the effect of small molecules, but possibly also protein modulators, on amyloid kinetics, which should become a standard in analysing amyloid kinetics in vitro.
Specific points
The different effects of drugs and CSF components on fragmentation, elongation and secondary nucleation are very interesting and should be discussed in more detail. At the moment, the discussion section does not really address any of the data in Figure 4. It looks like albumin specifically suppresses secondary nucleation, which is countered by VR16-09. What could be the molecular mechanism behind this?
EGCG curve looks superimposable to untreated control. Does EGCG remove part of the Abeta from the substrate pool or just reduce ThT binding?
Abeta fibril height is fairly small (average 5-6 nm) compared with previously reported data (8-10 nm). What is the authors’ interpretation in terms of the fibril structures? The change in height in response to the drugs is significant but also fairly small. What is the possible reason for these differences Do the authors propose that Abeta42 forms a different fibril structure in the presence of the drug? Does it alter the ratio of single vs. double stranded fibrils?
The authors hypothesize that CSF components increase the solubility of VR16-09. This should be tested. Can the drug be centrifuged out of solution?
It is not correct to report cell viability as the outcome of an MTT assay. MTT reports on the metabolic turnover. Numerous previous reports have shown that amyloid toxicity suppresses cell metabolism but does not necessarily kill the cells.
Minor points:
AFM images lack a height scale
The Abeta trace in presence of VR16-09 looks somewhat odd. Is VR16-09 fluorescent itself? The authors should include fluorescence traces of the drugs in a supplementary figure.
Reviewer 2 Report
The manuscript entitled “The major components of cerebrospinal fluid dictate the characteristics of inhibitors against amyloid-Beta amyloid aggregation” by Sakalauskas et al. is an important manuscript describing the effect of EGCG and VR16-09 on amyloid beta aggregation. The authors have pointed out that it is cerebrospinal fluid which decides whether a compound will act as inhibitor or not. The study highlights the role of cerebrospinal fluid in amyloid beta aggregation which is involved in Alzheimer’s disease and its inhibition. The manuscript has described the conditions which will have implications in discovery of inhibitors against amyloid related diseases. However, I have certain points which authors should take care before the manuscript may be accepted.
1. In the title “The major components of cerebrospinal fluid dictate the characteristics of inhibitors against amyloid-Beta amyloid aggregation”. Authors may think to change the title as “The major components of cerebrospinal fluid dictate the characteristics of inhibitors against amyloid-Beta Aggregation”
2. In the First paragraph of Introduction only Abeta 42 is mentioned while there should be mention of Abeta 40 as well (line 37).
3. Just before last paragraph in the Introduction section, there should be a brief description of inhibitors including EGCG and VR16-09.
4. In the Fig. 4, Y axis should be monomer equivalent concentration and not the fibril concentration. Also, authors should mention in the figure legend, which method (Like ThT fluorescence) has been used to follow the kinetics of Aβ1-42 aggregation.
5. In the line 213, only glutamine reference is given. Authors should include references of other amino acids on Abeta aggregation process, if any.
